# Validating the Use of Corifollitropin Alfa in Progestin-Primed Ovarian Stimulation Protocol on Normal and High Responders by Comparing with Conventional Antagonist Protocol: A Retrospective Study

**DOI:** 10.3390/life10060090

**Published:** 2020-06-21

**Authors:** Chen-Yu Huang, Guan-Yeu Chen, Miawh-Lirng Shieh, Hsin-Yang Li

**Affiliations:** 1Department of Obstetrics and Gynecology, Taipei Veterans General Hospital, 201, Shih-Pai Road Section 2, Taipei 112, Taiwan; eu.huang501@gmail.com (C.-Y.H.); shc1214clean@gmail.com (G.-Y.C.); doc3643h@yahoo.com.tw (M.-L.S.); 2Institute of Clinical Medicine, National Yang-Ming University, 155, Linong Street Section 2, Taipei 112, Taiwan; 3Division of Obstetrics and Gynecology, Faculty of Medicine, School of Medicine, National Yang-Ming University, 155, Linong Street Section 2, Taipei 112, Taiwan

**Keywords:** progestin-primed ovarian stimulation (PPOS), long-acting FSH, patient-friendly, antagonist

## Abstract

Our previous study showed a satisfactory reproductive outcome resulting from the patient-friendly ovarian stimulation protocol using long-acting follicle stimulation hormone (FSH) plus oral medroxyprogesterone acetate (MPA). The present retrospective study aims to compare the efficacy of the patient-friendly ovarian stimulation protocol with that of the antagonist protocol on normal and high responders aged between 24 and 39 years in a tertiary fertility center in Taiwan. To prevent premature luteinizing hormone (LH) surge, oral MPA was given to patients in group 1 (*n* = 57), whereas antagonist protocol was applied to group 2 (*n* = 53). Duration and dosage of stimulation, number of injections and visits before trigger, incidence of premature LH surge, number of oocytes retrieved, fertilization rate, cleavage rate, rate of good embryos available, incidence of ovarian hyperstimulation syndrome, cumulative clinical pregnancy rate and live birth rate per retrieval were compared between groups. We conclude that our patient-friendly ovarian stimulation protocol with MPA demonstrates satisfactory stimulation and reproductive outcomes that are comparable to those of an antagonist protocol.

## 1. Introduction

Conventional ovarian stimulation in assisted reproductive technology (ART) burdens patients with frequent injections including daily gonadotropins for follicle development and gonadotropin-releasing hormone (GnRH) analogues for premature luteinizing hormone (LH) surge prevention. In the past, GnRH agonist was the most popular method to prevent LH elevation and was named the long protocol owing to its lengthy course of daily injections [1]. Over the last decade, it has gradually been replaced by the GnRH antagonist protocol, which is characterized by immediate suppression of LH with fewer injections and lower risk of ovarian hyperstimulation syndrome (OHSS) [2]. Emerging from the demand for urgent fertility preservation in cancer patients [3], luteal phase stimulation has been adopted [4] in conjunction with freeze-all strategy, wherein the competence of the oocytes/embryos as well as the pregnancy and neonatal outcomes have already been validated on a large scale [5,6,7]. Stemming from the awareness of endogenous progesterone’s LH suppressive capability, oral progestogen has proven to be an effective alternative for LH rise prevention, namely progestin-primed ovarian stimulation (PPOS) [8,9,10]. Hence, multiple injections of GnRH analogues can be omitted.

Thanks to recombinant DNA technologies, one single shot of corifollitropin alfa can support follicle-stimulating activity equivalent to the circulating follicle stimulation hormone (FSH) level sufficient for multiple follicle growth throughout the first week [11] and thus replace seven daily FSH injections with compatible reproductive outcomes in conventional protocols [12]. This long-acting FSH has also been used in a random start fashion with antagonist protocol [13], but there is lack of investigation specific to luteal stimulation. To pursue a more patient-friendly way of ovarian stimulation in ART cycles, we combined PPOS with corifollitropin alfa and our proof-of-concept study demonstrates a competent reproductive outcome in normal/high responders with an ongoing pregnancy rate of 53.1% [14].

All of the previous studies regarding PPOS utilized daily injection of gonadotropins such as human menopausal gonadotropin (HMG), and most of them compared PPOS with short agonist protocol [8,9,15,16,17]. According to a worldwide survey of ART practices, more than three quarters of cycles utilized an antagonist protocol [18]; however, there is only limited literature for non-donor normal/high responders focused on the comparison between PPOS and antagonist protocol. A Chinese study for patients with polycystic ovarian syndrome showed a comparable ongoing pregnancy rate per transfer between PPOS group and antagonist-protocol group [19]. Another Iranian study showed a higher tendency of clinical pregnancy rate in the antagonist group as compared to the PPOS group despite no statistical significance [20]. One Japanese study for patients with normal ovarian reserve concluded similar ongoing pregnancy rates between PPOS and the antagonist group [21]. None of these evidences mentioned live birth rate and all of them used short-acting gonadotropin. Whether the use of long-acting FSH in PPOS has a comparable reproductive outcome with that of long-acting FSH in antagonist protocol is still unknown.

From December 2016, we began to administer PPOS with corifollitropin alfa to some patients undergoing ART in order to decrease the number of injections, while some patients received the standard antagonist protocol with corifollitropin alfa. The aim of our present study is to compare the reproductive outcomes, including live birth rate, of long-acting FSH (corifollitropin alfa) use in PPOS versus in an antagonist protocol.

## 2. Materials and Methods

### 2.1. Study Setting and Patients

A retrospective study was performed at the Center for Reproductive Medicine in Taipei Veterans General Hospital and undertaken by means of chart review. All patients were counseled and informed consent for the stimulation protocol and related procedures was provided by infertility specialists. Ethical approval for this study was obtained from the Institutional Review Board of Taipei Veterans General Hospital (2018-12-005BC). From December 2016 to August 2017, 235 of 623 cycles of ovarian stimulation for ART commenced with long-acting FSH (corifollitropin alfa) at early follicular phase, which was defined as either the first five days of menstruation cycle or basal serum estradiol less than 75 pg/mL along with basal serum progesterone below 1 ng/mL. Among these cycles, patients with one of the following conditions were excluded from the study: (1) age over 40 years, (2) antral follicle count (AFC) below 7, (3) basal FSH more than 10 IU/L, (4) a previous poor ovarian response (≤3 oocytes retrieved with a conventional stimulation protocol), (5) body mass index (BMI) above 30, (6) hypogonadotropic hypogonadism, and (7) uterine abnormalities (Figure 1). Ethical approval for this study was obtained from the Institutional Review Board of Taipei Veterans General Hospital (2018-12-005BC).

### 2.2. Treatment Protocol of Ovarian Stimulation and Oocyte Retrieval

Transvaginal ultrasound and serum hormone measurements (FSH, LH, estradiol (E2), and progesterone (P)) were performed on the starting day just before long-acting FSH (corifollitropin alfa) injection, of which the dosage was determined by the patient’s body weight (150 micrograms for >60 kg and 100 micrograms for ≤60 kg [22]). In cycles with PPOS (PPOS protocol, group 1), patients started to take oral medroxyprogesterone acetate (MPA) 5 mg BID from the day after long-acting FSH injection, which was published elsewhere [14]. In cycles with the antagonist protocol (antagonist protocol, group 2), daily cetrorelix 0.25 mg subcutaneous (sc) injection was initiated from the evening of stimulation day 5. Seven days after long-acting FSH injection, the folliculogenesis was monitored by transvaginal ultrasound along with serum hormone levels of E2, LH, and P. As long as at least three leading follicles reached above 17 mm in diameter, patients were triggered at night. If the follicle development was not adequate for trigger, additional HMG 150–225 IU/day would be given for days depending on the prediction according to the monitoring on stimulation day 8. If necessary, additional folliculometry would be performed every 2–3 days to evaluate whether the criterion for trigger was met. Patients in group 1 took the final tablet of MPA in the morning of the trigger day, whereas patients in group 2 received the final cetrorelix injection in the evening before the trigger day. Triggering was given by subcutaneous injection of triptorelin 0.2 mg with or without human chorionic gonadotropin (hCG) 1500–6500 IU, depending on the risk evaluation for early onset OHSS. A simplified illustration of the two treatment protocols is shown in Figure 2. Ovum pick-up was performed 34–38 h after triggering, followed by in vitro fertilization (IVF) and/or intracytoplasmic sperm injection (ICSI) according to the conditions of the sperm. All embryos were vitrified at pronuclear stage, or on day 2, 3, or 5 after oocyte retrieval depending on the number of fertilized oocytes available.

### 2.3. Endometrial Preparation for Frozen–Thawed Embryo Transfer

In frozen–thawed embryo transfer (FET) cycles, oral estradiol valerate 6 mg was given twice a day starting from menstrual cycle day 2 to 4. After oral estradiol valerate was taken for 10 to 14 days, endometrial thickness was measured by transvaginal sonography. In addition, serum hormone tests (E2, LH, P) were obtained to confirm that no spontaneous follicle growth and ovulation occurred. Once the endometrial thickness reached more than 7 mm, oral estradiol valerate supplement was continued and the patient was instructed to use vaginal micronized progesterone gel 90 mg twice a day and vaginal micronized progesterone soft capsules 400 mg every night until either ten weeks of gestation or confirmed pregnancy failure. The timing of thawing and transfer of frozen embryos was determined based on the stage they were vitrified and synchronized with the duration of progesterone exposure of the endometrium. Embryos frozen at the pronuclear stage were warmed after one day of vaginal progesterone treatment, followed by transfer one or two days later according to the number of viable embryos. Embryos frozen post-retrieval on day 2, 3 or 5 were thawed after 2, 3 or 5 days of luteal support, respectively, and were transferred on the same day. Serum β-hCG level was checked after two weeks of luteal support. If the serum β-hCG level was above 10 IU/L, transvaginal sonography was arranged three weeks later to verify for intrauterine pregnancy and fetal viability.

### 2.4. Outcome Measures

Demographic variables recorded for each patient included age, BMI, AFC, serum levels of basal FSH and LH, primary infertility, and the indication for IVF/ICSI treatments. Parameters for ovarian stimulation, oocytes, embryos, and pregnancy outcomes after FET include: duration and dosage of stimulation, the number of injections and visits before trigger, premature LH surge, the number of oocytes retrieved, fertilization rate, cleavage rate, the rate of good embryos available, OHSS, cumulative clinical pregnancy rate, and cumulative live birth rate per retrieval. In the study, peak E2 and LH levels on the trigger day were not necessarily measured due to freeze-all policy. A serum LH concentration over 10 IU/L or rising above twice the basal value before trigger was regarded as premature LH surge. Embryos growing to 2–4 cells one day after pronuclear stage with grade I or II morphology [23] were considered as good embryos. Cumulative clinical pregnancy rate was counted as pregnancies above 7 weeks with intrauterine fetal heartbeat detected by dividing by the number of retrieval cycles whose embryos were all transferred or confirmation of intrauterine pregnancies with active fetal heartbeat. Live birth was defined as delivery above 32 weeks of gestational age.

### 2.5. Statistical Analysis

Chi-squared test or two proportions test was used for comparisons of nominal variables between the two groups. Continuous parameters were analyzed with Student’s *t*-test for normal distributed data, presented as the mean ± standard deviation (SD), or Mann–Whitney U test for non-normal distributed data, presented as the median (minimum–maximum). The SPSS statistical package (version 24; SPSS Inc., Chicago, IL, USA) was used for analysis, and a *p*-value of less than 0.05 was considered to be statistically significant.

## 3. Results

In the study period, a total of 110 cycles of IVF/ICSI stimulated with corifollitropin alfa were included in this retrospective study. Among the 110 cycles, there were 57 cycles of PPOS (group 1) and 53 cycles of antagonist protocol (group 2) (Figure 1). Forty-five of the 57 cycles in group 1 were described in our previous proof-of-concept publication [14].

In regard to demographic data, age, BMI, AFC, serum basal FSH and LH levels, primary infertility, and the indication for IVF/ICSI treatments were comparable between the two groups. (Table 1).

There was no significant difference in the duration of stimulation, but the numbers of injections and visits before trigger were significantly lower in group 1. Total dosage of gonadotropin other than corifollitropin alfa consumed in group 1 was significantly lower than that of group 2. The number of oocytes retrieved, fertilization rate, cleavage rate, and good embryo rate on day 2 showed no significant difference between the two groups. None of the patients in either group developed either premature LH surge or OHSS (Table 2).

At time of submission, 54 patients in group 1 and 48 patients in group 2 either transferred all their cryopreserved embryos without pregnancy or achieved intrauterine pregnancy with good fetal heartbeat (Figure 1). In addition, 30 patients in group 1 and 28 patients in group 2 conceived through FET showing fetal viability at 7 weeks of gestation. Therefore, the cumulative clinical pregnancy rate per ovum pickup was 55.56% in group 1 and 58.33% in group 2, showing no statistically significant difference. Further, 28 patients in group 1 and 26 patients in group 2 delivered live birth without major sequelae, showing no significant difference in percentage between the groups (Table 3). Regarding the hormonal change in patients undergoing PPOS (group 1) versus antagonist protocol (group 2), the serum E2, LH, and P levels at basal status and at trigger were presented with the mean and SD in Figure 3. Due to freeze-all strategy, serum hormone tests on the trigger day were not necessarily performed; therefore, the data in Figure 3 came from only 14 patients in each group.

## 4. Discussion

The significance of our present work is that the use of long-acting FSH in PPOS has a comparable reproductive outcome with that of long-acting FSH in an antagonist protocol. To our knowledge, most publications to date regarding PPOS are based on HMG and whether MPA–PPOS is comparable with antagonist protocol in long-acting FSH application has not been reported. Thus, our study is the first to compare PPOS versus antagonist protocol with long-acting FSH use.

The major difference of our protocol compared with other PPOS regimens is the absence of exogenous LH effect during the first week of stimulation. Debate about the role of LH in ovarian stimulation is still continuing. Due to diverse results, the necessity of LH supplementation for patients undergoing IVF/ICSI cycles is still uncertain [24,25]. The updated Cochrane review showed no clear evidence of a difference between recombinant LH (rLH) combined with recombinant FSH (rFSH) and rFSH alone in live birth rates in spite of more ongoing pregnancies under LH supplementation, in which the benefits appeared to be more evident for low responders [26]. Hence, we do not propose using long-acting FSH plus MPA regimen in low responders. In the past, because of limited comprehension regarding the preventive effect of endogenous progesterone on premature LH surge, previous investigation about luteal start utilizing pure FSH [3,27,28] usually initiated daily GnRH antagonist from the first day of ovarian stimulation. In these regimens both GnRH antagonists and endogenous progesterone exerted suppressive influence and consequently resulted in much more profound pituitary suppression, thus requiring a higher dosage of FSH or longer stimulation duration. As for PPOS without GnRH antagonist, a group in China led by Kuang [8,10,29] demonstrated that MPA leads to stronger pituitary suppression in PPOS as compared with utrogestan and dydrogesterone, under which the LH values gradually declined in the first five days of MPA co-treatment. The proportion of women with profound pituitary suppression, which is defined as serum LH less than 1.0 IU/L on the trigger day, was 32% in the PPOS with daily MPA 10 mg commenced along with gonadotropin stimulation [30]. The aforementioned research used HMG for ovarian stimulation, so whether PPOS without LH supplementation is adequate remains unclear. The only study utilizing urinary FSH to compare with two brands of HMG in PPOS for normal responders [31] showed no differences in the number of retrieved oocytes, mature oocytes as well as fertilization, cleavage, embryo quality, and pregnancy outcomes among groups, even under subanalysis of patients with LH less than 0.68 IU/L on the trigger day. In our regimen, we arbitrarily started MPA supplement from the second day of gonadotropin stimulation, and the LH levels on stimulation day 8 were measured as 2.70 ± 1.84 IU/L in our previous study [14] and 2.85 ± 2.40 IU/L in the current investigation (data not shown). After stimulation day 8, HMG was added until trigger criteria were met. We demonstrated that the reproductive outcomes of long-acting FSH + MPA are satisfactory and comparable to those of long-acting FSH + antagonist. Certainly, further randomized controlled trials are needed.

Different progestin gives rise to different levels of pituitary suppression [10,29]. Previous research showed dydrogesterone 20 mg/d has less pituitary suppression than MPA 10 mg/d but more than utrogestan 100 mg/d. The pituitary suppression in utrogestan–PPOS is dose-dependent [32]; in contrast, MPA shows no difference in the proportion of profound pituitary suppression between 4 mg/d and 10 mg/d [30]. Despite various levels of pituitary suppression, there was no difference in the reproductive outcomes among various types and dosages of progestogen in the studies upon normal responders [10,29,32]. The difference might emerge in low responders stimulated with only FSH instead of HMG or FSH + LH. On the other hand, it seems that clomiphene co-administration is able to avoid LH oversuppression in PPOS, which resulted in an initial slight rise followed by a downward trend in LH, although the limited data came from normal responders [33] and high responders [34]. If we consider extending our long-acting FSH PPOS application to low responders, different progestin and clomiphene combinations might be necessary in PPOS with long-acting FSH where no exogenous LH is given in the first week.

Almost all previous relevant studies unanimously required a higher dosage of HMG with or without longer duration for stimulation in MPA–PPOS [8,16,35]. In our finding, there was lower gonadotropin consumption in PPOS and no significant difference in stimulation duration as compared to the antagonist protocol. There are two possible explanations: first, the comparison protocol in all of the aforementioned PPOS studies in the literature was a short protocol, which had flare-up effects that were absent in the antagonist protocol used in our comparison; second, we started MPA from the second day of gonadotropin stimulation, which is one day later than most of the PPOS protocols in the aforementioned studies and might lessen the pituitary suppression. Another retrospective cohort study comparing PPOS versus flexible antagonist protocol in the same donor aged between 23 and 29 years old administered MPA 10 mg/day from the seventh day of daily FSH stimulation or when the leading follicle reached 14 mm, whichever came first, and named this method as flexible progestin primed ovarian stimulation (fPPOS). Stimulation duration, gonadotropin consumption, and duration of GnRH antagonist/MPA administration were similar, and no premature ovulation occurred in either group. There were significantly more metaphase II oocytes in fPPOS cycles than in GnRH antagonist cycles. Recipients of fresh oocytes from fPPOS and GnRH antagonist cycles had similar cleavage, implantation, and live birth/ongoing pregnancy rates [36]. As for MPA–PPOS versus long protocol, which suppresses pituitary function much more than short and antagonist protocol, there was a randomized controlled trial including 257 patients aged less than 42 years with normal ovarian reserve. Not only the incidence of OHSS, but also the duration and dose of HMG stimulation were significantly higher in the long protocol group than those in the MPA–PPOS group. No significant difference was found in the number of oocytes retrieved and viable embryos as well as clinical pregnancy rate between PPOS and long protocols [37]. As for another progestogen, a retrospective study on patients with polycystic ovarian syndrome displayed a significant higher fertilization rate, viable embryo rate, and clinical pregnancy rate in utrogestan–PPOS group as compared to those receiving a short protocol under similar HMG consumption between groups [17]. Regarding dydrogesterone use in PPOS, there have been two randomized controlled trials: a Japanese one enrolling normal responders showed that the PPOS group took more HMG but ongoing pregnancy rate was comparable between PPOS and antagonist group [21]; the Iranian study for patients with polycystic ovarian syndrome showed a higher tendency of clinical pregnancy rate in the antagonist group as compared to PPOS group despite no statistical significance, and the dose of gonadotropin along with the duration of stimulation was similar between two groups [20].

According to previous investigation, the fertility potential of oocytes collected in the presence of exogenous progestogen, regardless of MPA [8,16], dydrogesterone [10,29,36], or utrogestan [9,17,38], is as competent as those collected via conventional stimulation protocols. An interesting big data study revealed no difference in the reproductive outcome of the oocytes induced in the presence or absence of a levonorgestrel-releasing intrauterine device [39]. Despite the highest concentration of levonorgestrel within the uterus, serum level of levonorgestrel varied between 134 and 191 pg/mL [40], thus offering further observation of the effect of prolonged progestin exposure on ovarian stimulation outcomes. In regard to endogenous progesterone, the reproductive outcomes of oocytes stimulated during luteal phase have already been qualified on a large scale [5,6,7].

As a novel stimulation method, currently there are no ideal candidates that benefit more from PPOS than other stimulation protocols. Progestins have been used for endometriosis therapy for decades [41], and, in theory, are able to balance the elevating estrogen level during controlled ovarian stimulation and to alleviate its related pain. Published data assessing PPOS in women with ovarian endometrioma, however, are scarce. A retrospective case-control study included normal responders with ovarian endometrioma proven either via surgery before ART or via aspiration during oocyte pickup. The results showed that higher rates of mature oocyte and high-quality embryo on day three as well as higher HMG dose were observed in MPA–PPOS groups as compared with the short protocol. Ongoing pregnancy rates were similar among groups [35]. Another prospective cohort study focused on fertility preservation for patients with ovarian endometrioma and altering ovarian reserve (AFC < 10 and/or anti-Mullerian hormone (AMH) level < 2). The numbers of oocytes retrieved and cryopreserved were similar between the PPOS group and antagonist group. In the cost-effectiveness analysis, the PPOS protocol was strongly dominant over the antagonist protocol [42].

‘Freeze-all’ is mandatory in PPOS. The leading benefit about FET is to eliminate the risk of late-onset OHSS without compromising the implantation rate [43]. There is no absolute predictor for OHSS and a presumed normal responder according to AMH or AFC still could have potential OHSS risk [44]. In our center, for that reason, the freeze-all policy is applied for both normal and high responders, and PPOS with long-acting FSH lightens their load in the physical/mental/economic aspects. The literature has described some regimens used to alleviate the stress from injections. Long-acting GnRH agonist depots was anticipated to be better than short-acting ones. In a retrospective study including more than 400 patients, higher OHSS risk and inferior reproductive outcomes were noted in the long-acting group [45], however there was no significant difference in another meta-analysis [46]. For better or worse, daily shots of gonadotropins were still needed in the aforementioned downregulation protocols. It was proposed that GnRH antagonist can be used occasionally only when serum LH exceeds 6 IU/L during ovarian stimulation started with long-acting FSH (corifollitropin alfa) [47]. In spite of reduced injections, this protocol needs frequent blood tests to monitor LH titers for timely initiation of antagonist injection, which is both bothersome and stressful. Despite close follow-up, it is still possible to miss some LH peaks. Without any prevention, the rate of premature LH surge during ovarian stimulation was 20–25% according to the literature [48]. Using our method, the first visit was arranged one week after long-acting FSH injection. A median of three injections and one visit before triggering is more convenient and friendly for patients.

Some limitations in our present work should be considered. First of all, this is a retrospective study, so further randomized controlled trial is warranted. Second, the sample number of our study is low, and a few of patients in each group did not complete their embryo transfer due to personal reasons, which made the sample size even smaller for live birth rate evaluation. Third, the conclusion is not generalizable either for other progestogens such as dydrogesterone and utrogestan, or for different GnRH antagonist agents such as ganirelix.

## 5. Conclusions

We believe that our protocol using long-acting FSH plus MPA is more patient friendly and significantly requires fewer visits and injections before trigger. Our results also displayed satisfactory stimulation and reproductive outcomes that are comparable to those of the antagonist protocol.

## Figures and Tables

**Figure 1 life-10-00090-f001:**
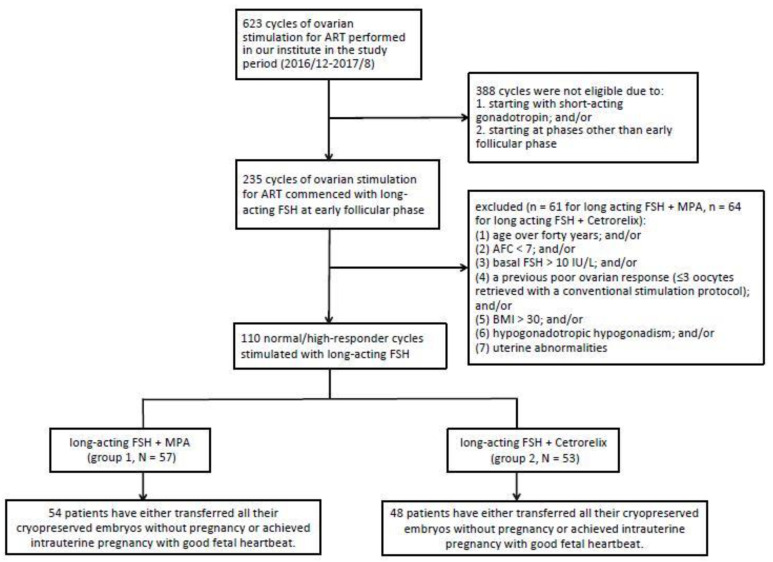
Flowchart of the study. Abbreviations: ART = assisted reproductive technology; FSH = follicle stimulation hormone; MPA = medroxyprogesterone acetate.

**Figure 2 life-10-00090-f002:**
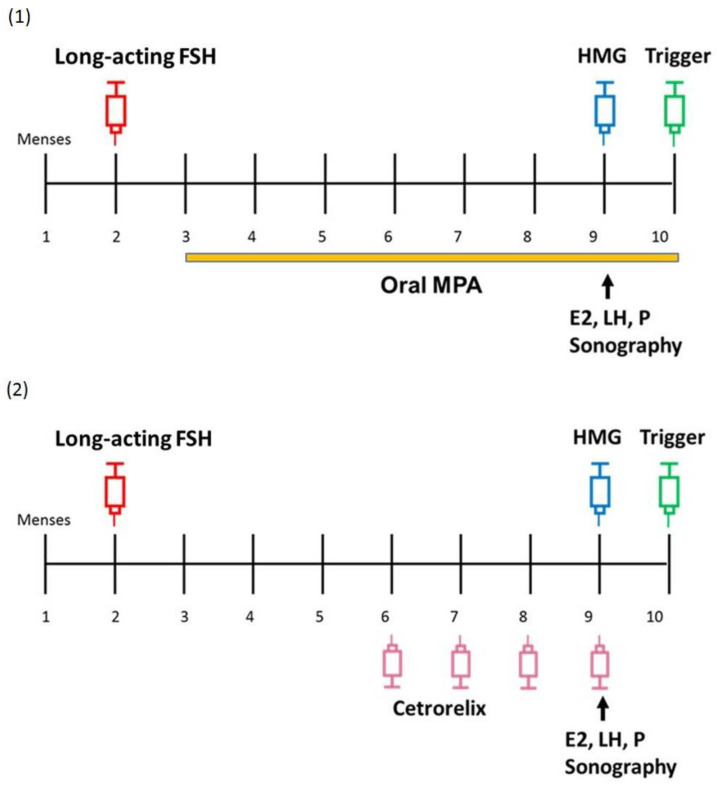
Simplified schematic descriptions of the two treatment protocols. Ovarian stimulation began with long-acting FSH (corifollitropin alfa) at early follicular phase. In cycles with PPOS. (Figure 2-(**1**), group 1), patients took medroxyprogesterone acetate (MPA) 5 mg twice a day from the day after long-acting FSH injection. In cycles with antagonist protocol. (Figure 2-(**2**), group 2), daily cetrorelix 0.25 mg sc was initiated from the evening of stimulation day 5. Seven days after long-acting FSH injection, the follicle development was monitored by transvaginal sonography as well as serum E2, LH and P measurements. As long as at least three leading follicles reached above 17 mm, patients were triggered at night. If the trigger criterion was not met, additional HMG 150–225 IU/day would be administered for days depending on the prediction according to the measurement on stimulation day 8. If needed, additional follicle monitoring would be performed every 2–3 days before trigger. Patients in group 1 took the final tablet of MPA in the morning of the trigger day, whereas patients in group 2 received the final cetrorelix in the evening before the trigger day. Abbreviations: E2 = estradiol; FSH = follicle stimulation hormone; HMG = human menopausal gonadotropin; LH = luteinizing hormone; MPA = medroxyprogesterone acetate; P = progesterone.

**Figure 3 life-10-00090-f003:**
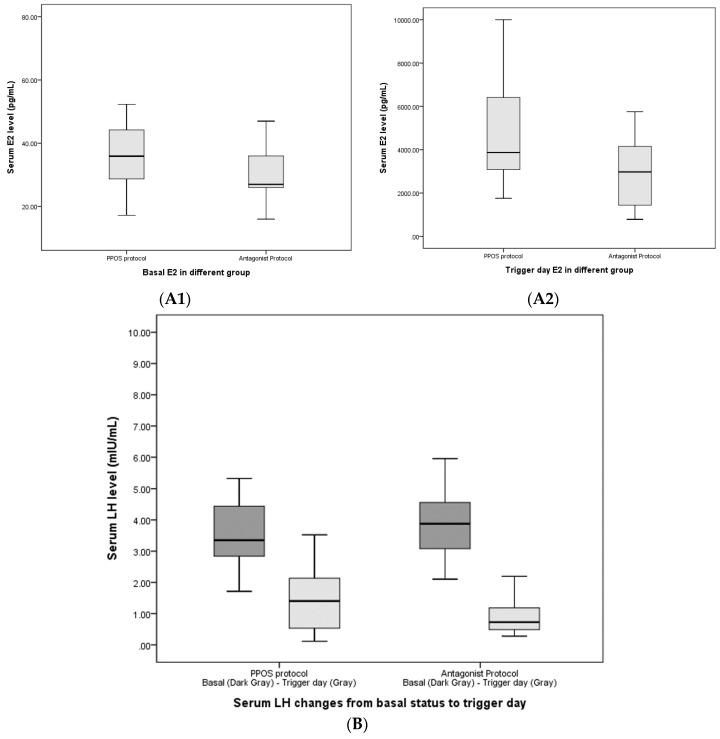
Hormonal changes in patients undergoing progestin-primed ovarian stimulation (PPOS) (group 1) versus antagonist protocol (group 2). (**A1**) Serum estradiol (E2) comparison in basal status; (**A2**) Serum E2 comparison at trigger day. (**B**) Serum LH changes from basal status to trigger day. (**C**) Serum progesterone changes from basal status to trigger day.

**Table 1 life-10-00090-t001:** Demographic characteristics.

Characteristics	Long-Acting FSH + MPA (Group 1, *n* = 57)	Long-Acting FSH + Cetrorelix (Group 2, *n* = 53)	*p*-Value
Age (Mean ± SD)	34.8 ± 2.73	34.9 ± 3.41	0.865
BMI (Mean ± SD)	21.54 ± 2.55	21.75 ± 2.97	0.693
BMI ≤ 24 (*n*, %) ^§^	45, 78.95%	39, 73.58%	0.519
BMI > 24 (*n*, %) ^§^	12, 21.05%	14, 26.42%	
AFC (median (minimum–maximum)) ^#^	16 (6–64)	14 (4–62)	0.202
Basal FSH (IU/L) (mean ± SD)	6.37 ± 1.15	6.48 ± 1.92	0.733
Basal LH (IU/L) (mean ± SD)	4.37 ± 2.29	3.90 ± 1.71	0.235
Primary infertility (*n*, %) ^§^	35, 61.40%	26, 49.06%	0.193
The indication for IVF/ICSI treatments ^§^			0.283
Male factor (*n*, %)	20, 35.08%	16, 30.19%	
Tubal factor (*n*, %)	9, 15.78%	15, 28.30%	
Other (*n*, %)	28, 49.12%	22, 41.51%	

^#^ Mann–Whitney U test; ^§^ Chi-squared test. Abbreviations: SD = standard deviation; BMI = body mass index; AFC = antral follicle count; FSH = follicle stimulation hormone; IU = international unit; LH = luteinizing hormone; IVF = in vitro fertilization; ICSI = intracytoplasmic sperm injection.

**Table 2 life-10-00090-t002:** Stimulation characteristics and data of oocytes and embryos.

Variable	Long-Acting FSH + MPA (Group 1, *n* = 57)	Long-Acting FSH + Cetrorelix (Group 2, *n* = 53)	*p*-Value
No. of injections before trigger (median (minimum–maximum)) ^#^	3 (2–9)	10 (5–18)	<0.001 *
No. of visits between long-acting FSH injection and trigger (median (minimum–maximum)) ^#^	1 (1–2)	2 (1–3)	<0.001 *
Duration of stimulation (days) (median (minimum–maximum)) ^#^	9 (8–15)	9 (6–13)	0.724
Total dosage of gonadotropin other than long-acting FSH (IU) (median (minimum–maximum)) ^#^	450 (225–1800)	600 (150–1800)	<0.001 *
Premature LH surge	0	0	-
No. of oocytes retrieved (median (minimum–maximum)) ^#^	12 (1–43)	13 (2–39)	0.679
Fertilization rate (%) (median (minimum–maximum)) ^#^	83.33 (26.67–100)	73.87 (0–100)	0.070
Cleavage rate (%) (median (minimum–maximum)) ^#^	100 (65–100)	90 (25–100)	0.203
D2 good embryo rate (%) (median (minimum–maximum)) ^#^	66.67 (0–100)	58.33 (0–100)	0.804
Incidence of ovarian hyperstimulation syndrome (OHSS)	0	0	-

^#^ Mann–Whitney U test; * *p* < 0.05.

**Table 3 life-10-00090-t003:** Pregnancy outcomes of frozen–thawed embryo transfer (FET).

	Long-Acting FSH + MPA (Group 1, *n* = 54)	Long-Acting FSH + Cetrorelix (Group 2, *n* = 48)	Difference (95% Confidence Interval)	*p*-Value
Cumulative clinical pregnancy rate with good fetal heartbeat ^※^	30 (55.56%)	28 (58.33%)	2.77% (−16.00% ~ 21.15%)	0.779
Cumulative live birth rate at the time of submission ^※^	28 (51.85%)	26 (54.17%)	2.32% (−16.53% ~ 20.89%)	0.816

Data are in numbers (%), ^※^ two proportions test.

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
