# Peer review of "Validating the Use of Corifollitropin Alfa in Progestin-Primed Ovarian Stimulation Protocol on Normal and High Responders by Comparing with Conventional Antagonist Protocol: A Retrospective Study"

_life, 2020, doi:10.3390/life10060090_

Round 1
Reviewer 1 Report
The authors of this retrospective study compared the duration of stimulation, number of shots and visits before trigger, incidence of premature LH surge, number of oocytes retrieved, fertilization rate, cleavage rate, rate of good embryos available, incidence of ovarian hyperstimulation syndrome, cumulative clinical pregnancy rate and live birth rate per retrieval between two groups of normal and high responders treated by IVF/ICSI . One group was stimulated with the use of an antagonist protocol and the other with a progestin-controlled protocol using medroxyprogesterone acetate (MPA). No differences in biological and clinical outcomes were observed between the two groups, but the MPA protocol required much fewer injections and visits. The paper is well designed and well written, and the statictical methods are adequate. I recommend acceptance after some revision along the lines below.
- There is a time overlap between this study and another study, describing the protocol but not including a control group, by the same authors (Reprod. Biol. Endocrinol. 16:18, 2018). The authors should state whether some patients in the MPA protocol were included in both studies.
- The authors should state clearly what was the rationale for treating patients with each of the two protocols. Were some patients treated first with the GnRHa protocol and then with the MPA protocol?
- There are at least 3 studies, 2 of them RCTs using progestin-primed cycles. The two RCTs used dydrogesterone for priming in one group and an antagonist protocol in the other. (Iwami et al., Arch. Gynecol. Obstet. 2018. https://doi.org/10.1007/s00404-018-4856-8) (Eftekhar et al., Int. J. Reprod. Biomed. 17, 671-676, 2019). The third study used Utrogestan for progestin priming and compared outcomes with the short agonist protocol (Zhu et al., Medicine 95,28(e4193), 2016). The authors should discuss these papers and point out how their data contribute to the current debate as to the choice of the optimal progestin for primiming
Author Response
- There is a time overlap between this study and another study, describing the protocol but not including a control group, by the same authors (Reprod. Biol. Endocrinol. 16:18, 2018). The authors should state whether some patients in the MPA protocol were included in both studies.
Thank you for the suggestion. We have added the relevant statement in the Result section. (Please refer to page 14, line 197-198.)
- The authors should state clearly what was the rationale for treating patients with each of the two protocols. Were some patients treated first with the GnRHa protocol and then with the MPA protocol?
Our rationale to use PPOS with long-acting FSH is simply to decrease the number of injections. From December 2016, we began to randomly apply PPOS with corifollitropin alfa to some patients undergoing ART, and other patients received conventional antagonist protocol. We added the aforementioned statement in the Introduction section. (Please refer to page 5, line 90-92) There were no patients treated with antagonist protocol before the MPA protocol.
- There are at least 3 studies, 2 of them RCTs using progestin-primed cycles. The two RCTs used dydrogesterone for priming in one group and an antagonist protocol in the other. (Iwami et al., Arch. Gynecol. Obstet. 2018. https://doi.org/10.1007/s00404-018-4856-8) (Eftekhar et al., Int. J. Reprod. Biomed. 17, 671-676, 2019). The third study used Utrogestan for progestin priming and compared outcomes with the short agonist protocol (Zhu et al., Medicine 95,28(e4193), 2016). The authors should discuss these papers and point out how their data contribute to the current debate as to the choice of the optimal progestin for primiming
Thank you for the suggestion. We added these reference (Ref. 17, 20, 21) to our discussion (please refer to page 24-25, line 340-351) and, together with our original text, we had a thorough discussion about different progestogens. (Please refer to page 22-25, line 299-351.) However, there are no sufficient evidences yet to conclude the best progestogen for priming.
Reviewer 2 Report
I revised the manuscript entitled “Validation of an extremely patient-friendly progestin-primed ovarian stimulation protocol in normal and high responders by comparing with conventional antagonist protocol: a retrospective study” (Manuscript Number: life-808183).
The authors performed a retrospective study aimed to compare the efficacy of the extremely patient-friendly ovarian stimulation protocol with that of antagonist protocol on normal and high responder women aged between 24 and 39 years.
In my honest opinion, the topic is interesting enough to attract the readers’ attention.
Nevertheless, the methodology description should be improved, and conclusions are only partially supported by the data analysis. The authors should clarify some points and improve the discussion discussing better study limitations of the study that are not evidenced in the discussion.
In general, the Manuscript may benefit from several major revisions, as suggested below:
- All the text needs a language revision by a native English speaker, in order to improve its readability.
- I would suggest providing a better background for the use of progestogens to avoid LH surge during ovarian stimulation. The previous study of the authors should be substituted by a brief summary of available evidence on the use of progestogens during controlled ovarian stimulation. What we do not know that this study aimed to answer?
- Being a retrospective study method should report the source of data. The number of identified patients should be reported in the results.
- I would suggest improving the introduction better clarifying what is known and what is unknown about the adoption of progestogens to prevent LH surge during ovarian stimulation. Which gap in the literature this study is aimed to cover?
- Inclusion and exclusion criteria should be improved.
- Results, I would suggest reporting the total number of IVF/ICSI cycles performed in the study period, the number of cycles excluded with progesterone use, and the same with GnRH use. A flow-chart could be helpful.
- Line 182. What does “the number of injections” refer to? If it includes the GnRH antagonist used in the second group, this number does not make sense. Clearly, if you adopt an injective drug the number of injections will be higher. It is more interesting if the total doses of gonadotropins were different.
- Methods and results. The number of visits should be compared with non-parametric tests, it is not a continuous variable and the median instead of the mean may better represent the actual number of visits. 1.4 or 1.9 visits are difficult to interpret.
- Figure 2. I would suggest using dot-plot with box and whiskers graphs.
- Lines 218-219. This statement is not supported by study results. I would suggest summarizing the study results, the reduced number of injections is an expected result cause by the avoided use of GnRH analogs. Therefore, the real results are only the comparable clinical pregnancy rate. The reduced number of injections is an advantage of progesterone use.
- The adjective “extremely patient-friendly” for the reported protocol should be supported by evidence. This study did not assess the experience of patients. I would suggest better discussing this point and avoiding to report it in the title.
- I would suggest significantly shortening the discussion focusing mainly on the study results and aim related to the use of progesterone for the control of LH surge during ovarian stimulation. Many parts can be shortened or avoided.
- Study limits are not discussed.
- One of the studies who adopted the use of progestogens for ovarian stimulation tested the use in women with ovarian endometriomas, I would suggest better discussing this possible advantage of progestogens. (PMID: 31755673; PMID: 28931865)
Author Response
- All the text needs a language revision by a native English speaker, in order to improve its readability.
We have had our revised manuscript checked by a native English speaking colleague, Jen-Yu Tseng, which was also mentioned in the Acknowledgments section. (Please refer to page 29, line 431.)
- I would suggest providing a better background for the use of progestogens to avoid LH surge during ovarian stimulation. The previous study of the authors should be substituted by a brief summary of available evidence on the use of progestogens during controlled ovarian stimulation. What we do not know that this study aimed to answer?
- I would suggest improving the introduction better clarifying what is known and what is unknown about the adoption of progestogens to prevent LH surge during ovarian stimulation. Which gap in the literature this study is aimed to cover?
(Suggestion 2. & 4.)
Thank you for your suggestion. In the Introduction section, we have shortened the statement about our previous study and reinforced the background about what is known (previous researches on comparison with conventional protocol) and what is unknown (no studies about long-acting FSH application in PPOS and lack of data about live birth rate in PPOS vs antagonist protocol). Please refer to page 4, line 75-89.
- Being a retrospective study method should report the source of data. The number of identified patients should be reported in the results.
- Inclusion and exclusion criteria should be improved.
- Results, I would suggest reporting the total number of IVF/ICSI cycles performed in the study period, the number of cycles excluded with progesterone use, and the same with GnRH use. A flow-chart could be helpful.
(Suggestion 3. & 5. & 6.)
Thank you for the excellent suggestion. We have made a flowchart (Figure 1) to make the source of data along with the number of identified patients as well as inclusion and exclusion criteria more clear: There were 623 cycles of ovarian stimulation for ART performed in our institute in the study period (2016/12-2017/8), among which 235 cycles commenced with long-acting FSH at early follicular phase and was therefore eligible. 125 cycles with one of the following conditions were excluded from the study: (1) age over forty years, (2) antral follicle count (AFC) below 7, (3) basal FSH more than 10 IU/L, (4) a previous poor ovarian response (≤3 oocytes retrieved with a conventional stimulation protocol), (5) body mass index (BMI) above 30, (6) hypogonadotropic hypogonadism, and (7) uterine abnormalities. Then 110 normal/high-responder cycles stimulated with long-acting FSH were included, among which fifty-seven cycles received PPOS (group 1) and fifty-three cycles underwent antagonist protocol (group 2).
- Line 182. What does “the number of injections” refer to? If it includes the GnRH antagonist used in the second group, this number does not make sense. Clearly, if you adopt an injective drug the number of injections will be higher. It is more interesting if the total doses of gonadotropins were different.
- Methods and results. The number of visits should be compared with non-parametric tests, it is not a continuous variable and the median instead of the mean may better represent the actual number of visits. 1.4 or 1.9 visits are difficult to interpret.
(Suggestion 7. & 8.)
Thank you for the excellent suggestion. We re-analyzed some variables, such as the number of visits and injections, with non-parametric tests (# Mann-Whitney U test), and added the data about the total doses of gonadotropins other than corifollitropin alfa. Please see Methods, and Table 1 and Table 2 in the Result section.
- Figure 2. I would suggest using dot-plot with box and whiskers graphs.
In the revised version, it became Figure 3. Thank you for the suggestion and we revised the figure as dot-plot with box and whiskers graphs. (Please see Figure 3)
- Lines 218-219. This statement is not supported by study results. I would suggest summarizing the study results, the reduced number of injections is an expected result cause by the avoided use of GnRH analogs. Therefore, the real results are only the comparable clinical pregnancy rate. The reduced number of injections is an advantage of progesterone use.
We have revised the relevant statement. (Please refer to page 19, line 248-250.)
- The adjective “extremely patient-friendly” for the reported protocol should be supported by evidence. This study did not assess the experience of patients. I would suggest better discussing this point and avoiding to report it in the title.
Thank you for the suggestion. We have revised the title.
- I would suggest significantly shortening the discussion focusing mainly on the study results and aim related to the use of progesterone for the control of LH surge during ovarian stimulation. Many parts can be shortened or avoided.
We have revised the discussion according to your suggestion.
- Study limits are not discussed.
Thank you for the important suggestion. We have added the discussion about the limitation of our study. Please refer to page 27-28, line 406-412.
- One of the studies who adopted the use of progestogens for ovarian stimulation tested the use in women with ovarian endometriomas, I would suggest better discussing this possible advantage of progestogens. (PMID: 31755673; PMID: 28931865)
Thank you for the suggestion. We added a paragraph about the relevant discussion with the citation of literature (Ref. 35, 41) you offered. Please refer to page 25-26, line 365-379.
Round 2
Reviewer 2 Report
I revised the revised version of the manuscript entitled “Validating the use of corifollitropin alfa in progestin-primed ovarian stimulation protocol on normal and high responders by comparing with conventional antagonist protocol: a retrospective study” (Manuscript Number: life-808183).
I appreciated the manuscript improvement; however, the statistical analysis needs further revision and improvement. In the present form is not correct.
In table 2, the authors report the mean and SD for proportions, which is not correct. This raises concerns about how actually ware compared the cumulative incidences between the two groups. Which test did the author use? Only t-test and Mann-Whitney? They cannot be used for categorical variables (proportions).
Moreover, I would suggest revising the description in the methods section. Statistical analysis should be improved reporting all the used tests and how the assumptions of t-test were tested.
What does “*” mean in tables? I would suggest improving the legends of tables clarifying for each p-value the adopted test and any abbreviation or symbol reported.
Finally, regarding at least the primary outcome, I would suggest reporting the difference in proportion and the 95% confidence interval. Based on the value, the authors will be able to support with variable strength that no difference is present between the two compared protocols.
Author Response
In Round 2 revision, we used the "Track Changes" function in Microsoft Word as the editor suggested, so that changes are easily visible. (The Round 1 revisions in the manuscript were marked up [yellow-colored highlighted with red text] WITHOUT track changes according to the suggestion of editors)
Thank you very much for the suggestion about further revision and improvement on the statistical analysis, and we have revised point by point on the basis of your comments:
- revising the description in the methods section--Statistical analysis should be improved reporting all the used tests and how the assumptions of t-test were tested.
We used t-test under the assumption of normal data distribution. We have revised our description in the Statistical Analysis section. Please refer to page 13, line 186-192.
- In table 2, the authors report the mean and SD for proportions, which is not correct. This raises concerns about how actually ware compared the cumulative incidences between the two groups. Which test did the author use? Only t-test and Mann-Whitney? They cannot be used for categorical variables (proportions).
Sorry for the misleading description in Table 2. In Table 2, the 3 parameters presented in % are actually continuous variables (not proportions), which are defined as below:
Fertilization rate = the number of fertilized oocytes/ the number of oocytes retrieved per retrieval
Cleavage rate = number of 2PN zygotes cleaved / number of 2PN zygotes per retrieval
D2 good embryo rate = number of day 2 good embryo (defined as line 178)/ number of 2PN zygotes per retrieval
We presented them as averages for each group via t-test in our original version, but, thank you for the reminding, we re-analyzed them with Mann-Whitney U test because of their non-normal distribution. Please refer to Table 2 in revised version. (line 216)
- What does “*” mean in tables? I would suggest improving the legends of tables clarifying for each p-value the adopted test and any abbreviation or symbol reported.
Thank you for the important reminder. We have added the footnote in tables about *, which means p<0.05. (Please refer to line 218 and the footnotes below each table)
- Finally, regarding at least the primary outcome, I would suggest reporting the difference in proportion and the 95% confidence interval. Based on the value, the authors will be able to support with variable strength that no difference is present between the two compared protocols.
Thank you for the excellent suggestion. We have added the difference in proportion and the 95% confidence interval. Please refer to Table 3 in the revised version. (line 236)
Round 3
Reviewer 2 Report
No further comments.